# Functional-genomic analysis reveals intraspecies diversification of antiviral receptor transporter proteins in *Xenopus laevis*

**Ian N. Boys**, **Katrina B. Mar**, **John W. Schoggins***

Department of Microbiology, University of Texas Southwestern Medical Center, Dallas, Texas, United States of America

* john.schoggins@utsouthwestern.edu

**Data Availability Statement:** Xenopus laevis RNA-seq data are available from the NCBI through the Gene Expression Omnibus (accession number

## Abstract

The Receptor Transporter Protein (RTP) family is present in most, if not all jawed vertebrates. Most of our knowledge of this protein family comes from studies on mammalian RTPs, which are multi-function proteins that regulate cell-surface G-protein coupled receptor levels, influence olfactory system development, regulate immune signaling, and directly inhibit viral infection. However, mammals comprise less than one-tenth of extant vertebrate species, and our knowledge about the expression, function, and evolution of non-mammalian RTPs is limited. Here, we explore the evolutionary history of RTPs in vertebrates. We identify signatures of positive selection in many vertebrate RTP clades and characterize multiple, independent expansions of the RTP family outside of what has been described in mammals. We find a striking expansion of RTPs in the African clawed frog, *Xenopus laevis*, with 11 RTPs in this species as opposed to 1 to 4 in most other species. RNA sequencing revealed that most *X. laevis* RTPs are upregulated following immune stimulation. In functional assays, we demonstrate that at least three of these *X. laevis* RTPs inhibit infection by RNA viruses, suggesting that RTP homologs may serve as antiviral effectors outside of Mammalia.

## Author summary

Receptor Transporter Proteins (RTPs) are a family of proteins which, in mammals, are involved in diverse physiological processes such as olfaction and innate immunity. Less, however, is known about non-mammalian RTPs. In this study, we investigate the evolutionary history of RTPs and identify antiviral RTPs in the African clawed frog, *Xenopus laevis*.

## Introduction

Receptor Transporter Proteins (RTPs) were initially identified as regulators of chemosensory receptors [1]. In mammals, there are four RTPs: RTP1s, RTP2, RTP3, and RTP4. Since their

GSE166455). Other relevant data are within the manuscript and its Supporting Information Files.

**Funding:** This study was in part supported by NIH grants to J.W.S. (AI117922 and AI158124), I.N.B and K.B.M. (T32 AI005284), as well as grants to J. W.S. from the UTSW Endowed Scholars Program, the Rita Allen Foundation, and The Welch Foundation (I-2013-20190330). This study was additionally supported by an NSF GRFP (2016217834) to I.N.B. J.W.S. holds an Investigators in the Pathogenesis of Infectious Disease Award from the Burroughs Wellcome Fund. Any opinion, findings, and conclusions or recommendations expressed in this material are those of the authors(s) and do not necessarily reflect the views of funding agencies. The funders had no role in study design, data collection and analysis, decision to publish, or preparation of the manuscript.

**Competing interests:** The authors have declared that no competing interests exist.

discovery, RTPs have been implicated in diverse cellular and physiological processes. Mammalian RTPs have been shown to regulate the localization of diverse G-coupled protein receptors [2,3], and, in the case of RTP1 and RTP2, influence the development of the olfactory system [4]. Uniquely among mammalian RTPs, recent work has uncovered roles for RTP4 in the interferon (IFN) response, a key component of antiviral immunity. RTP4 is both an antiviral effector that restricts infection by RNA viruses of the family *Flaviviridae* [5] and a negative regulator of TBK1-mediated signaling pathways [6]. While its precise antiviral mechanism is unknown, RTP4 binds replicating viral RNA via its N-terminal zinc finger domain. This association alters the behavior of viral replication machinery, leading to a decreased association between the viral polymerase (NS5) and the viral helicase (NS3), altered binding of viral polymerase to viral RNA, and reduced replication of viral RNA [5]. We recently found evidence of a genetic arms race between mammalian RTP4 and flaviviruses, in which viruses have driven unique adaptations in RTP4 in diverse mammals [5]. This finding prompted us to ask whether similar RTP-virus evolutionary conflicts have arisen in other vertebrates.

The RTP family is ancient, dating back to the origin of jawed vertebrates; however, little is known about the function of these evolutionarily distant vertebrate RTPs. Interestingly, RTP homologs from several species of fish are induced by IFN [7–9] and one homolog has been implicated as a resistance allele to viral disease in Asian sea bass [10]. This is suggestive of a role for non-mammalian RTPs in the innate immune response to viruses. Here, we take an evolution-guided approach to identify non-mammalian RTPs with antiviral properties by characterizing the evolutionary trajectories of multiple RTP clades. Unexpectedly, we encountered a remarkable expansion of RTPs within the African clawed frog, *Xenopus laevis*. We used RNA sequencing to identify pathogen-associated molecular pattern (PAMP)-induced *X. laevis* RTPs and screened for their ability to inhibit viral infection relative to representative RTPs from other species. Using this functional-genomic approach, we identified multiple antiviral *X. laevis* RTPs with unique viral specificities, indicating that antiviral innovation is a common property of vertebrate RTPs.

## Results

Non-mammalian RTPs, such as fish "*RTP3*", have been previously described as orthologs–not homologs–of mammalian RTPs. A maximum-likelihood tree generated from an alignment of 303 vertebrate RTPs (File A in S1 File) shows that non-mammalian RTPs generally form distinct clades among related species, consistent with a model in which intragenomic RTP expansions have typically resulted from more recent duplication events rather than ancient duplications (Fig 1A). Anamniote (fish and amphibian) RTPs form a phylogenetically-distinct cluster, yet there is some ambiguity regarding the relationships between sauropsid (reptile and bird) and mammalian RTPs. While mammalian RTPs distinctly cluster in pairs (*RTP1* with *RTP2* and *RTP3* with *RTP4*), bird RTPs cluster with the RTP3/4 clade, whereas reptile RTPs cluster variably with the RTP1/2 clade and the RTP3/4 clade, and one from the bearded dragon clusters weakly with anamniote RTPs (Fig 1A). It is unclear whether this is indicative of a divergence of the progenitor RTPs of mammalian RTP1/2 and RTP3/4 before or after the divergence of sauropsids and mammals.

Jawed fish have between one and three RTPs, some of which cluster most closely within species. For example, the two Atlantic herring RTPs are more-closely related to each other than to those found in other fish, suggestive of repeat gene duplications over evolutionary time (Fig 1B). Concerted evolution, a phenomenon in which homologous recombination events and other processes result in increased similarity between paralogous genes within one species than between orthologs from related species, could partially underly this observation.

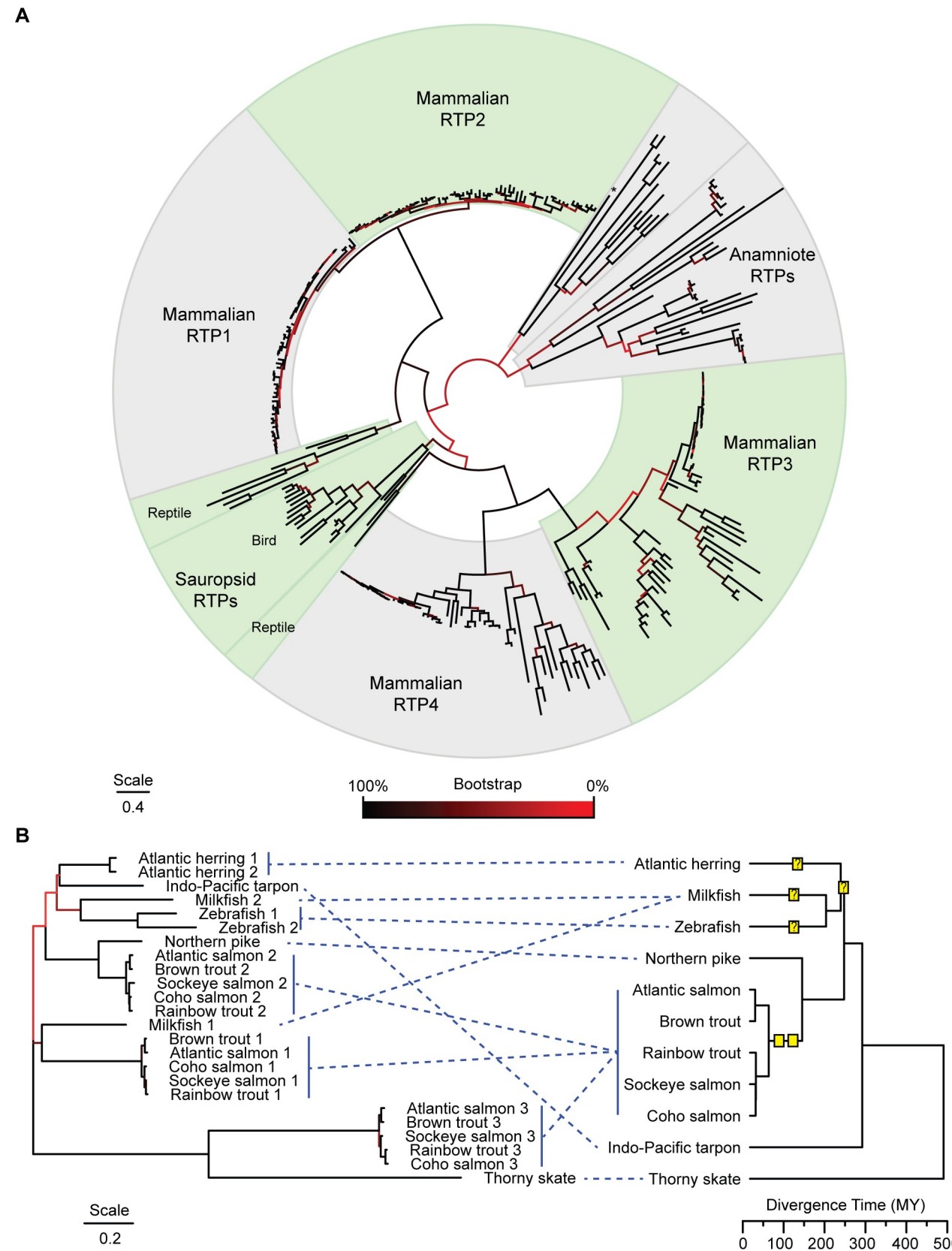

**Fig 1. Evolutionary survey of vertebrate RTPs.** A) A Maximum Likelihood tree for 303 vertebrate RTPs was inferred by using the Maximum Likelihood method and Tamura-Nei model. The tree with the highest log likelihood (-99002.44) is shown. 500 bootstrap replicates were performed to test robustness of the ML tree, and branches are colored by percentage of bootstrap replicates which reflect this topology. A discrete Gamma distribution was used to model evolutionary rate differences among sites (5 categories (+G, parameter = 0.8741)). The tree is drawn to scale, with branch lengths measured in the number of substitutions per site. Asterix (*) denotes the bearded dragon RTP that clusters with anamniote RTPs. Tree with species and gene names included is available in File B in S1 File. B) Left: A Maximum Likelihood tree for 24 representative fish RTPs was inferred by using the Maximum Likelihood method and Tamura-Nei model. The tree with the highest log likelihood (-6168.52) is shown. 500 bootstrap replicates were performed to test robustness of the ML tree, and branches are colored by percentage of bootstrap replicates which reflect this topology. A discrete Gamma distribution was used to model evolutionary rate differences among sites (5 categories (+G, parameter = 2.2558)). The tree is drawn to scale, with branch lengths measured in the number of substitutions per site. Right: A TimeTree describing the evolutionary relationships between the different fish species. Yellow squares denote inferred duplication events. Question marks denote ambiguous duplication events.

However, we found only limited evidence of recombination in fish RTPs. An analysis of fish RTPs using a Genetic Algorithm for Recombination Detection (GARD) [11] indicated one likely recombination breakpoint ($\Delta AIC_c = 14.96$) at nucleotide 383 in our alignment of fish RTPs (File C in S1 File). Maximum-likelihood trees for gene fragments on either side of the recombination breakpoint produced a similar topology (S1A Fig), consistent with a model in which recent duplications, not gene conversion events, have resulted in the topology observed in Fig 1B. Further supporting this conclusion, using GENECONV [12], we found among fish RTPs evidence of only one possible partial gene conversion (Milkfish RTP 1 outer p = 0.0160), which corresponded to an indel-rich region in fish RTPs (position 73–90 in fish alignment found in File C in S1 File). Additional support for recent duplication events was found within the Euteleostei, where salmoniforme (an order which contains salmon and trout) genomes contain three RTPs, and the closely-related Northern pike has only one, indicative of two duplication events following their divergence.

Despite these recurrent duplications, RTPs are generally syntenic (Fig 2A). There are two examples of non-syntenic RTPs in the genomes of species we analyzed: one in salmon and one in the house mouse (Fig 2A). Our analysis revealed a curious discrepancy in domain architecture between representative anamniote (fish and amphibian) and amniote (reptile, bird, and mammals) RTPs. All amniote RTPs contain transmembrane (TM) domains, RTPs from fish lack a TM domain, and RTPs from amphibians are variable (Fig 2A). Importantly, studies have found that the TM domain of mammalian RTPs is dispensable for their receptor trafficking [13] and antiviral [5] roles. Thus, the lack of a TM domain in many anamniote RTPs does not exclude them from serving either a receptor trafficking or antiviral function.

We also noted that the RTP family is remarkably expanded in the African clawed frog, *Xenopus laevis*; its tetraploid genome contains eleven RTPs, more than any other species for which genomic information is available (Fig 2A). The evolutionary history of *X. laevis* is unique among known vertebrates. Approximately 34 million years ago, an ancestral species diverged, forming two *X. laevis* progenitors. Around 17 million years later, the divergent diploid ancestors hybridized and both subgenomes were maintained in their tetraploid descendants, which over the next 17 million years evolved to become the *X. laevis* of the present day [14]. Previous work has identified disproportionate pseudogenization on the smaller of the two *X. laevis* genomes (S) when compared to the larger (L) genome [15,16], which is likely a result of an energy-saving adaptation to limit genetic redundancy. We found no evidence of pseudogenization of *X. laevis* RTPs, and a "free-ratio" analysis using Phylogenetic Analysis by Maximum Likelihood (PAML) [17] found evidence of positive selection (ratio of nonsynonymous to synonymous codon substitutions, or dN/dS, > 1) in multiple branches of *X. laevis* RTPs (Fig 2B). We further used adaptive Branch-Site Random Effects Likelihood (aBSREL) [18], a more-stringent test, to assess selection in the most-expanded syntenic family of *X. laevis* RTPs, here termed the "epsilon" family. This analysis detected evidence of episodic positive

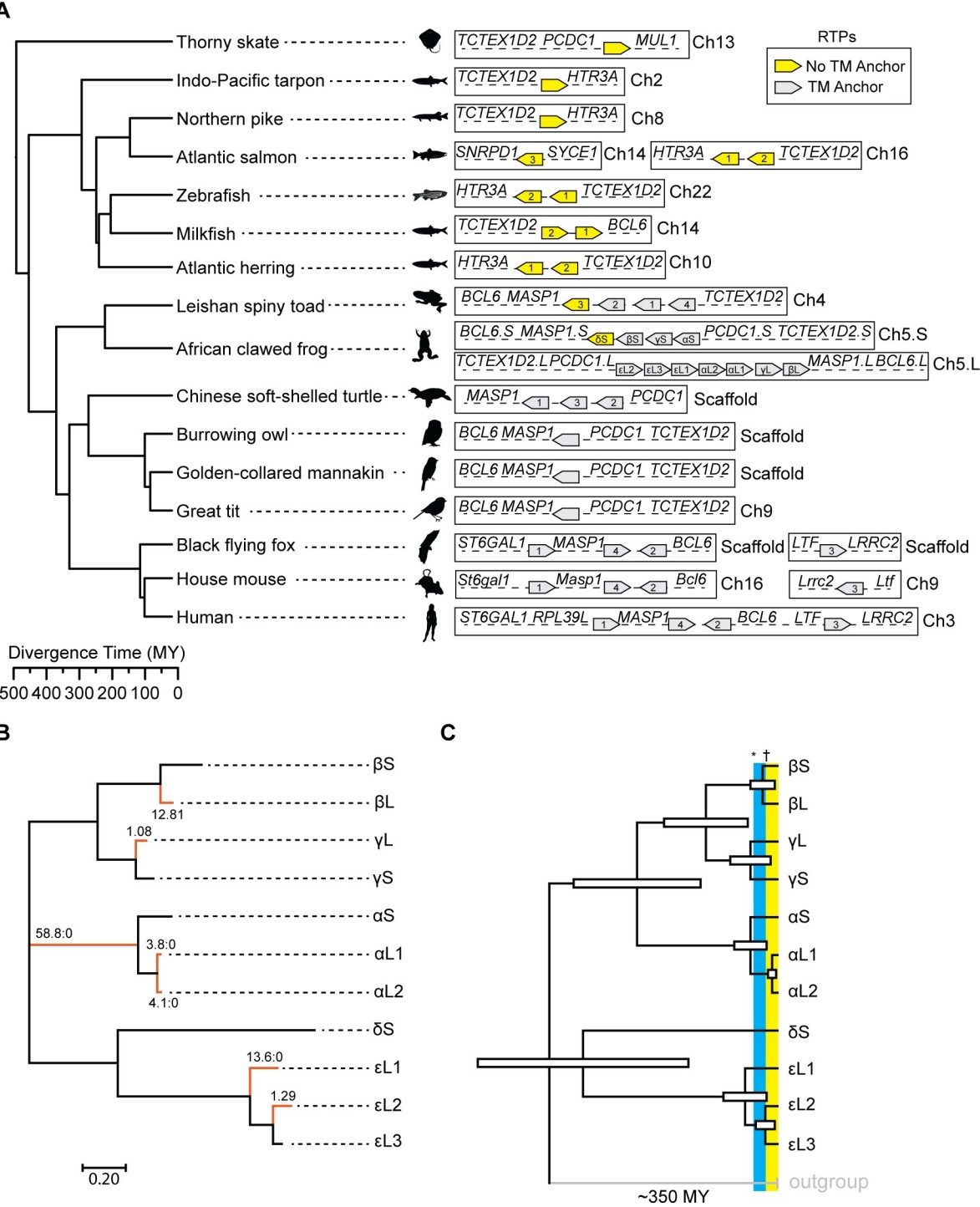

**Fig 2. RTP locus synteny and evolutionary analysis of *X. laevis* RTPs.** A) Left: A TimeTree [51] of representative RTP-containing vertebrate species. Right: chromosomal arrangement of RTPs. Common proximal genes and chromosome numbers, when available, are indicated. Numbering of RTP genes, when indicated, matches the numbering used in Fig 1B, File A in S1 File, and S4 Table. Scaffold denotes contigs that are not mapped to chromosomes. B) Free-ratio analysis of *X. laevis* RTPs. Annotated values indicate the dN/dS of each branch. Branches with dN/dS > 1 are marked in red. C) A TimeTree for *X. laevis* RTPs inferred using the Reltime method [52,53] and the Tamura-Nei model [54]. Divergence times were estimated by calibration by designating mammalian RTP4 sequences as an outgroup. The estimated log likelihood value is -9335.49. A discrete Gamma distribution was used to model evolutionary rate differences among sites (5 categories (+G, parameter = 2.9731)). The rate variation model allowed for some sites to be evolutionarily invariable ([+I], 6.22% sites). White rectangles denote 95% CI of divergence times. Asterix (*) indicates estimated L and S genome divergence time. Dagger (†) indicates estimated L and S genome hybridization time.

selection for two out of three genes ($\varepsilon$L1 p = 0.0018, $\varepsilon$L2 p = 0.0017), supporting our PAML results and highlighting the continued adaptation of *X. laevis* RTPs over evolutionary time (S1 Table). Together, these results are consistent with continued functional relevance despite the potential genetic redundancy that results from tetraploidy. Since more *X. laevis* RTPs exist than can be described by allotetraploidy, we next assessed whether any duplications occurred during the separation of the S and L genomes or following their rehybridization. We used a set of mammalian RTP4s to calibrate a time tree and estimate the divergence times of *X. laevis* RTPs (Fig 2C). Some nodes are ambiguous, such as that between the S genome "delta" RTP and the L genome "epsilon" RTPs. Conversely, the "beta", "gamma", and "alpha" gene families all had S and L paralog divergence times that roughly overlapped with the known divergence time between the S and L genomes (34 million years ago). Interestingly, we observed that the two L genome "alpha" RTPs, which have undergone episodic diversification (Fig 2B), diverged post-hybridization, further suggesting that RTP family expansion has been selected for in *X. laevis*.

Positive selection is a hallmark of genes which are involved in host defense against pathogens [19]. We previously found that mammalian *RTP4* is a rapidly evolving antiviral effector which inhibits infection by RNA viruses from the family *Flaviviridae* [5]. We used Fast, Unconstrained Bayesian AppRoximation for Inferring Selection (FUBAR) [20], Branch-site Unrestricted Statistical Test for Episodic Diversification (BUSTED) [21], and PAML to assess whether other RTPs are evolving under positive selection. Using gene-level models (PAML M7 vs M8 and M8 vs M8a in addition to BUSTED), we found evidence of positive selection in several non-mammalian RTP clades. At least two of the three models found evidence of positive selection in the bird RTP, as well as in two of the three salmon RTPs (Table 1). Similar to *RTP4*, mammalian *RTP3* exhibits signatures of positive selection in bats, rodents, and primates. The third salmon RTP, as well as mammalian *RTP1* and *RTP2*, did not exhibit robust

**Table 1. Evolutionary analysis of select vertebrate RTP families.**

|  | n sequences | *P* (M7 vs M8) | *P* (M8 vs M8a) | *P* BUSTED | NS BEB $\omega > 1$ | *FUBAR* (n positive) | *FUBAR* (n purifying) | dN/dS (gene) |
|---|---|---|---|---|---|---|---|---|
| Bird | 18 | *0.007962 | *0.033198 | 0.354 | 1 | 0 | 30 | 0.364 |
| Salmoniform 1 | 7 | *3.45E-04 | *6.98E-05 | 0.221 | 3 | 8 | 6 | 0.852 |
| Salmoniform 2 | 5 | *7.22E-01 | *4.23E-01 | 0.500 | 0 | 1 | 6 | 0.684 |
| Salmoniform 3 | 6 | 0.56841 | 0.29089 | *0.031 | 0 | 2 | 9 | 0.255 |
| *X. laevis* | 11 | *0.019284 | *0.00691 | 0.500 | 1 | 1 | 17 | 0.436 |
| Primate 1 | 25 | 0.999725 | 0.3054 | 0.500 | 0 | 0 | 38 | 0.096 |
| Primate 2 | 26 | *0.036599 | 0.1958 | 0.500 | 0 | 2 | 33 | 0.148 |
| Primate 3 | 27 | *4.28E-05 | *1.21E-05 | 0.147 | 4 | 9 | 11 | 0.704 |
| Primate 4 | 28 | *0.000482 | *0.000371 | 0.136 | 3 | 4 | 17 | 0.630 |
| Rodent 1 | 23 | *5.26E-05 | 0.3585 | 0.500 | 0 | 0 | 152 | 0.040 |
| Rodent 2 | 24 | 0.366 | *0.03493 | 0.497 | 0 | 4 | 108 | 0.067 |
| Rodent 3 | 22 | *0.000163 | *0.000265 | *0.001 | 1 | 1 | 57 | 0.504 |
| Rodent 4 | 16 | *1.42E-09 | *5.05E-07 | *0.000 | 9 | 7 | 13 | 0.630 |
| Bat 1 | 9 | 0.9995 | 0.4176 | 0.500 | 0 | 0 | 58 | 0.065 |
| Bat 2 | 11 | 0.9997 | 0.113 | 0.500 | 0 | 0 | 54 | 0.081 |
| Bat 3 | 11 | *1.97E-05 | *8.17E-06 | *0.000 | 6 | 11 | 17 | 0.698 |
| Bat 4 | 9 | *6.19E-12 | *1.36E-12 | *0.000 | 16 | 26 | 6 | 1.114 |

For PAML analyses: Positively-selected sites are those with an M8 BEB >95%. Both M7 vs M8 and M8a vs M8 tests compare site models which allow positive selection (M8) and those that do not (M7, M8a). Test results which indicate gene-wide signatures of positive selection are denoted with an asterisk. Underlying data can be found in File D in S1 File.

or consistent signatures of positive selection. Complementing these analyses, site-specific (PAML M8 Bayes Empirical Bayes and FUBAR) tests identified positively-selected residues in many of the same genes which displayed a gene-wide signature of selection.

Since we observed signatures of positive selection for mammalian *RTP3* (Table 1) which could be consistent with a role in host defense, we assessed the antiviral potential of mammalian RTP paralogs in mice, humans, and the black flying fox, by ectopic expression in a human hepatoma cell line (Huh7.5) that is permissive to infection by diverse viruses. Black flying fox RTP3 (paRTP3) was the only RTP other than the RTP4 homologs that inhibited viral infection (S2A Fig). paRTP3, which expressed at high levels (S2B Fig), was broadly antiviral, inhibiting the diverse RNA viruses yellow fever virus (YFV, a flavivirus), hepatitis C virus (HCV, a hepacivirus), Venezuelan equine encephalitis virus (VEEV, an alphavirus), and coxsackievirus B3 (CVB, a picornavirus). This contrasts with antiviral mammalian RTP4s, which primarily inhibit closely related flaviviruses and hepaciviruses (S2A Fig) [5]. This lack of viral specificity was paired with qualitative alterations in cellular morphology such as swelling and increased granularity (S2C Fig) as well as differences in cellular metabolism, as assayed by WST-1 conversion (S2D Fig). Broad antiviral phenotypes and altered cellular properties are consistent with viral inhibition via an indirect, host-dependent process, suggesting, albeit not concluding, that paRTP3 is not a bona fide antiviral effector.

The gene family expansions noted in Figs 1 and 2, as well as evidence of recurrent positive selection (Table 1) in non-mammalian RTPs, led us to hypothesize that some non-mammalian RTPs may have antiviral functions. Antiviral genes are often upregulated during infection, either directly as a result of the host cell sensing pathogen-associated molecular patterns (PAMPs), or as a result of IFN signaling [22]. Indeed, antiviral mammalian RTP4 is induced by IFN, as are several fish RTPs [7–9]. Additionally, one RTP from the Eastern newt, *Notophthalmus viridescens*, is upregulated during fungal infection, suggesting an immune role in amphibians [23]. The inducibility of *X. laevis* RTPs, however, has not been explored. *X. laevis* A6 kidney cells are IFN-competent and respond to a variety of immune agonists, including poly(I:C), a viral PAMP mimic [24]. We confirmed that Mx-family GTPases, a canonical family of interferon-stimulated genes (ISGs), were induced following transfection of A6 cells with poly(I:C) (Fig 3A). We next performed RNA-sequencing on poly(I:C)-transfected A6 cells over a time course. We identified n = 614, n = 859, and n = 647 upregulated (p < 0.05, fold-change > 4) genes 6, 12, and 24 hours post-transfection, respectively (Fig 3B and S2 Table). Temporal subcluster analysis of gene expression kinetics revealed ten clusters (Fig 3C), including two distinct clusters (SC4 and SC5) which were characterized by rapid and sustained induction following poly(I:C) transfection (Fig 3D). While poly(I:C) is less specific than IFN at inducing ISGs, these subclusters were rich in canonical ISGs, such as *IFIT*s, *RSAD2*, *CH25H*, and *MX2*, but also contained many RTPs (Fig 3E and S2 Table). Indeed, ten of the eleven *X. laevis* RTPs (all except for εL3) were significantly-induced at one or more time points (Fig 3F).

Unexpectedly, we also observed that two transcripts in SC5 are endogenized adintovirus polymerase genes (S2 Table). *Adintoviridae* is a recently-proposed [25] viral family that was identified in metagenomic data. Adintoviruses are characterized by adenovirus-like virion proteins that are associated with a retrovirus-like integrase. Endogenized gene fragments of adintoviruses are found in a variety of animal genomes. While endogenized viral DNA polymerases from the unrelated *Hepadnaviridae* have been described in birds [26,27], this is to our knowledge the first description of an immune agonist-responsive endogenized viral polymerase. Such regulation raises the prospect that amphibian species may have co-opted such proteins for a role in antiviral immunity.

While many genes with no known direct effector function are upregulated upon pathogen sensing, the inducibility of multiple *X. laevis* RTPs (Fig 3F), coupled with their robust

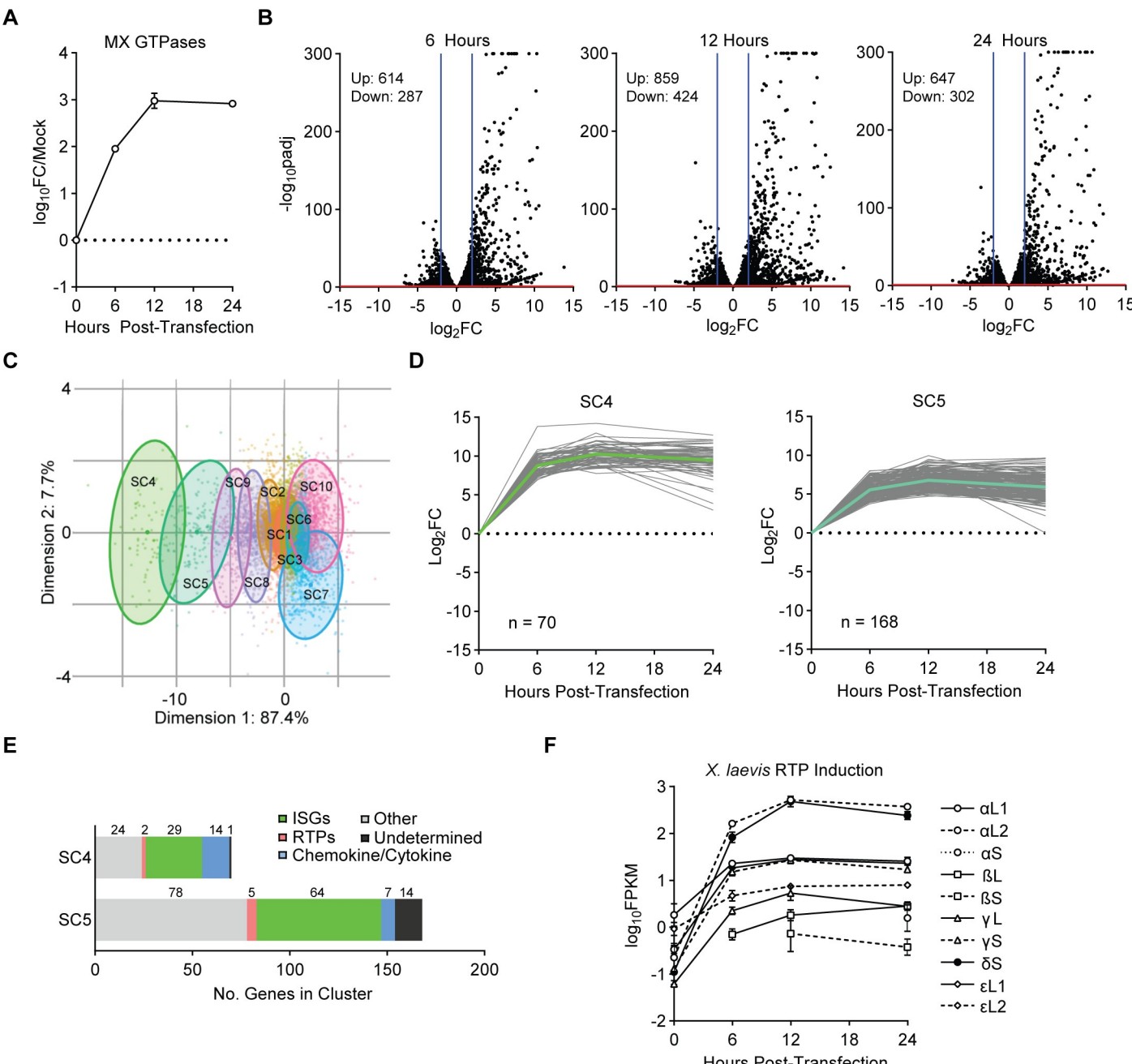

**Fig 3. Identification of poly(I:C)-induced *Xenopus laevis* RTPs.** A) qPCR was used to assess MX-family GTPase induction following transfection of A6 cells with poly(I:C) over a time course. As designed, primers are capable of detecting all *X. laevis* MX-family transcripts. Points represent the mean ± SD of n = 3 biological replicates. B) Volcano plots for A6 cells treated with poly(I:C) for 6, 12, or 24 hours, relative to mock transfection. Differentially-expressed genes (padj < 0.05, fold change > 4) are indicated. Data are derived from three biological replicates. C) PCA analysis of gene expression kinetics. Ten subclusters were identified. D) Kinetic profiles of top-induced, ISG-rich clusters (SC4 and SC5). Lines represent individual genes. Colored/bolded line indicates mean profile for the cluster. E) Annotation of genes in SC4 and SC5. ISGs annotated per inducibility in other datasets, as detailed in methods. Other: non-ISGs with clear annotation. Undetermined: insufficient homology to determine identify of transcript. F) Induction of RTP family members following poly(I:C) transfection. Plotted points indicate when significant enrichment over background was observed.

signatures of positive selection (Fig 2B), led us to hypothesize that *X. laevis* RTPs may represent an expanded antiviral protein family with similarities to the IFN-inducible, antiviral mammalian RTP4. To explore the antiviral potential of *X. laevis* RTPs and other non-

mammalian RTPs, we expressed in Huh7.5 cells HA-epitope tagged, codon-optimized RTP homologs from two birds (great tit and golden-collared manakin), an IFN-induced RTP [8] from the Atlantic salmon (salmoniform RTP 2), and all eleven homologs from *X. laevis*. We included black flying fox RTP4 (*Pteropus alecto* RTP4, paRTP4), a potent inhibitor of most flaviviruses, as a positive control for antiviral activity [5]. We performed a screen with a panel of representative positive-sense single-stranded RNA viruses, including yellow fever virus (YFV, a flavivirus), human coronavirus OC43 (HCoV-OC43), coxsackievirus B3 (CVB, a picornavirus), and Venezuelan equine encephalitis virus (VEEV, an alphavirus), as well as the negative-sense single-stranded RNA virus vesicular stomatitis virus (VSV, a rhabdovirus). Expression levels were variable, as epitope-tagged RTPα homologs, RTPδS, great tit RTP, and the Atlantic salmon RTP were undetectable by western blot (S3A Fig) while some RTPs such as RTPε and RTPγ homologs were expressed at moderate levels compared to higher-expressing RTPs such as RTPβL and paRTP4. This discrepancy across RTPs suggests that a lack of antiviral activity for any orthologs may be linked to a lack of efficient protein expression. Despite this limitation, we found that two homologs from *X. laevis* reduced infection by at least 50% relative to a vector control: *X. laevis* RTPγS restricted the flavivirus YFV, and *X. laevis* RTPαL1 restricted the picornavirus CVB (Figs 4A and S3B).

Mammalian RTP4 orthologs exhibit distinct, mosaic antiviral specificities towards members of the *Flaviviridae* [5]. We therefore screened the YFV-inhibiting RTPγS and its counterpart on the L genome, RTPγL, for their ability to inhibit other members of the *Flaviviridae* when ectopically expressed. We found that, in addition to YFV, RTPγS modestly inhibited the flaviviruses dengue virus (DENV) and Entebbe bat virus (ENTV) but did not inhibit other viruses (Fig 4B and 4C). Unexpectedly, while RTPγL did not inhibit any flaviviruses, it restricted the related hepacivirus hepatitis C virus (HCV), which was not inhibited by RTPγS (Fig 4B and 4C).

We next sought to determine which phase of the flavivirus replication cycle is targeted by RTPγS. After viral entry, the replication cycle can broadly be divided into three phases: initial translation of the incoming viral genome, genome replication, and virion assembly/egress. Mammalian RTP4 binds viral RNA and inhibits flavivirus genome amplification, a later step in the replication cycle [5]. To assess which step is affected by RTPγS, we used a minimal, replication-competent, Renilla luciferase (RLuc)-expressing YFV RNA referred to as a 'subgenomic replicon' which bypasses viral entry and differentiates early (translation) and late (replication) steps. We found that RTPγS inhibits replication but does not inhibit primary translation, consistent with what is known for mammalian RTP4 (Fig 4D)[5].

We next assessed whether RTPαL1, which inhibited the picornavirus CVB, or the related RTPαL2 or RTPαS, inhibited other picornaviruses when ectopically expressed. We found that, in addition to CVB, RTPαL1 inhibited Mengovirus (MenV, also called EMCV), and to a lesser degree poliovirus (PV), but did not significantly inhibit echovirus E11 (Fig 4E and 4F). RTPαL2 and RTPαS, however, did not significantly inhibit any of the viruses tested.

We previously determined that the 3CXXC zinc finger domain (ZFD) of RTP4 is its core antiviral domain, and we identified conserved cysteine residues that, when mutated to alanine, led to a loss of antiviral function [5]. We used site-directed mutagenesis to disrupt one of these motifs in RTPαL1 and RTPγS (S3C Fig) and found that RTPs bearing these ZFD-disrupting mutations (*ZFD) no longer inhibited MenV and YFV, respectively (Fig 4G and 4H). Importantly, RTPγS*ZFD expressed to higher levels than WT RTPγS, suggesting that the loss of function was not the result of protein instability (S3D Fig). Our inability to detect RTPαL1 by western blot (S3A Fig), however, precluded our ability to assess whether the loss of antiviral function of RTPαL1*ZFD is independent of protein expression. Finally, we used cross-linking immunoprecipitation (CLIP) paired with qPCR to assess whether antiviral *X. laevis* RTPs bind

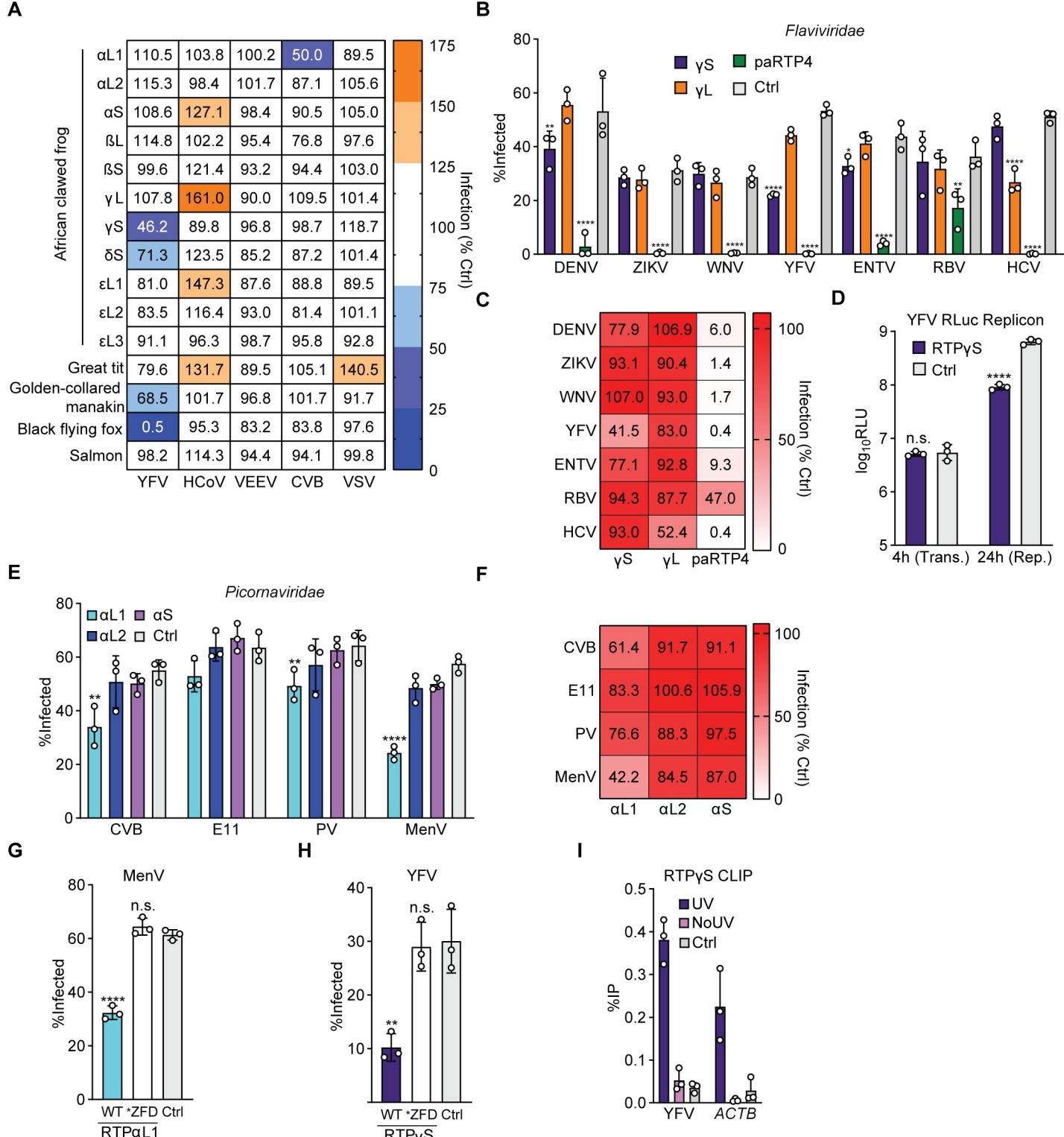

**Fig 4. Identification of antiviral *Xenopus laevis* RTPs.** A) Huh7.5 cells ectopically expressing the indicated RTPs or a vector control were infected with a panel of RNA viruses at an MOI of 0.5 to 1. Cells were harvested after the completion of approximately one replication cycle and percent infection was assessed by flow cytometry. Cells represent the mean of n = 2 biological replicates, normalized to control. Raw data: S2B Fig. B) Huh7.5 cells ectopically expressing the indicated RTPs or a vector control were infected with a panel of flaviviruses at an MOI of 0.5 to 1. Cells were harvested after the completion of approximately one replication cycle and percent infection was assessed by flow cytometry. Bars represent the mean ± SD of n = 3 biological replicates. C) Data from Fig 4B, represented as percent of control. D) Huh7.5

cells ectopically expressing either RTPγS or a vector control were transfected with a YFV-RLuc replicon and protein production was assessed by luminometry. Bars represent the mean ± SD of n = 3 biological replicates. E) Huh7.5 cells ectopically expressing the indicated RTPs or a vector control were infected with a panel of picornaviruses at an MOI of 0.5 to 1. Cells were harvested after the completion of approximately one replication cycle and percent infection was assessed by flow cytometry. Bars represent the mean ± SD of n = 3 biological replicates. F) Data from Fig 4E, represented as percent of control. G) Huh7.5 cells expressing HA.RTPγS, HA.RTPγS.C151A, or a vector control were infected with YFV-17D for 24 hours. Cells were harvested after the completion of approximately one replication cycle and percent infection was assessed by flow cytometry. Bars represent the mean ± SD of n = 3 biological replicates. H) Huh7.5 cells expressing HA.RTPαL1, HA.RTPαL1. C154A, or a vector control were infected with MenV for 6 hours. Cells were harvested after the completion of approximately one replication cycle and percent infection was assessed by flow cytometry. Bars represent the mean ± SD of n = 3 biological replicates. I) Huh7.5 cells expressing HA.RTPγS or a vector control were infected with YFV (MOI of 2) for 24h. CLIP-qPCR identified RNA bound by HA.RTPγS. UV: UV crosslinked HA.RTPγS cells. NoUV: non-crosslinked HA.RTPγS cells. Vector: UV crosslinked vector control cells. Bars: mean ± SD of n = 3 biological replicates.

viral RNA during infection. Perhaps because of low expression levels of RTPαL1, we only observed modest yet insignificant enhancement of MenV RNA bound to RTPαL1 compared to a vector control (S3E Fig). RTPγS, however, robustly bound YFV RNA, as well as host RNA, during infection (Fig 4I). Paired with the replication-specific phenotype (Fig 4D), the robust association of RTPγS with viral RNA indicates that antiviral *Xenopus* RTPs may share a similar antiviral mechanism of action with mammalian RTP4.

## Discussion

In the present study, we extend functional and genetic characterization of Receptor Transporter Proteins beyond Mammalia. It has previously been shown that RTP4 in mammals is an IFN-induced antiviral effector [5,28], and that several RTPs in fish are likewise upregulated by IFN [8,9]. However, mechanistic studies have thus far not been performed to assess the antiviral potential of non-mammalian RTPs. We identify signatures of positive selection in many vertebrate RTP clades, and characterize multiple, independent expansions of the RTP family outside of what was previously described in mammals. We identify a marked expansion of RTPs in the tetraploid African clawed frog, *X. laevis*, and find that these RTPs have continued to adapt and diversify following expansion. By performing what is to our knowledge the first deep sequencing-based analysis of PAMP-induced genes in *X. laevis*, we additionally find that 10 of these *Xenopus* RTPs are induced by the innate immune response to a mimic of viral infection. Critically, we functionally characterize these *Xenopus* RTPs and find that several inhibit viral infection. Like mammalian RTP4 orthologs, these antiviral *X. laevis* RTPs exhibit distinct antiviral specificity, inhibiting only certain viruses from certain viral families. One *X. laevis* RTP, RTPγS, inhibits viral replication and directly binds viral RNA during infection, which is similar to the antiviral mechanism of mammalian RTP4. Mammalian RTP4 is engaged in a host-pathogen molecular "arms race" [5], and if RTPs in non-mammalian species are bona-fide antiviral effectors, they may have influenced RNA virus evolution over the course of nearly half a billion years.

A key limitation of the present study is its reliance upon artificial combinations of non-mammalian RTPs with viruses that are known to predominantly infect mammals. While this allowed us to identify RTPs with antiviral activity, we predict that these RTPs may be evolutionarily honed to inhibit viruses that they are more likely to encounter in nature. Indeed, in our studies regarding mammalian RTP4, we encountered orthologs, such as human RTP4 and mouse Rtp4, which seem to have narrow antiviral specificity towards distinct subsets of pathogenic mammalian viruses [5]. The identification and phenotypic screening of amphibian RNA viruses that are relatives of those in the present work may provide better models for the study of antiviral *Xenopus* RTPs, which may yield candidate genes for knockout studies to investigate their physiological relevance in innate immunity. Indeed, a novel family of picornaviruses has recently been characterized in amphibians [29,30], and metagenomic studies have found that amphibians are unappreciated hosts to RNA viruses that are related to those which infect

mammals [31]. In addition to identifying RNA viruses that naturally infect amphibians and could therefore serve as models for RTP knockout studies, an important focus of future work will be the development of tissue culture-based models that permit the study of *Xenopus* RTPs and other antiviral effectors in a native cellular background. Of note, ectopic gene expression in A6 cells is inefficient [32], so it would be helpful to test other *Xenopus* cell lines as possible models.

A next step after our expression profiling of *Xenopus* RTPs in cell culture would be to assess *in vivo* expression patterns of RTPs following viral infection or immune stimulation. The expression of *X. laevis* RTPs at steady state [33,34] may however provide hints of their potential immune function as well as those outside of cell-intrinsic immunity. Many RTPs (RTPγL, RTPγS, RTPαL2, RTPβL) are expressed at low to intermediate levels in most tissues, which is perhaps expected for genes that are involved in antiviral immunity. By contrast, some RTPs are expressed highly in some tissues but not in others, which could be indicative of either tissue-specific antiviral roles, or tissue-specific developmental or housekeeping functions. RTPαS is highly expressed in the ovaries and during embryonic development, and RTPaL1 is expressed highly in the heart and lung. All three members of the RTPε family and RTPδS are predominantly expressed in the intestine, and RTPβS is expressed at the highest levels in the brain and lung. While we limited the scope of our mechanistic studies to the identification of antiviral roles for *Xenopus* RTPs, such expression patterns may warrant their study in other contexts. For instance, mammalian RTPs have been extensively studied in the context of olfaction, and *X. laevis* is an important model organism for the study of the olfactory system, particularly from a developmental perspective [35]. It is thus possible that *Xenopus laevis* may provide an attractive model for studying the impact of RTP evolution on the olfactory system.

## Materials and methods

### Cell culture

Huh7.5 and HEK-293T cells were maintained in DMEM supplemented with 10% FBS. *STAT1*$^{-/-}$ fibroblasts were maintained in RPMI supplemented in 10% FBS. BHK-21J cells were maintained in MEM supplemented with 10% FBS. A6 cells (ATCC CCL-102) were maintained in NCTC 109 media supplemented with 1x L-Glutamine, 15% ddH2O, and 10% FBS. All cells with the exception of A6 cells were cultured at 37°C in 5% CO2. A6 cells were maintained at 28°C in 5% CO2. Stable Huh7.5 cell lines were maintained by passaging in the presence of 4μg/mL puromycin.

### Viruses

The generation and propagation of the following viruses have been previously described: YFV17D-Venus, HCV genotype 2a intragenotypic chimera expressing Ypet (HCV-Ypet), CVB-GFP, and WNV-GFP [36,37]. VSV-GFP was produced by passaging in BHK cells. A ZIKV MR766-GFP infectious clone (kindly provided by Matt Evans, Icahn School of Medicine at Mount Sinai) was used to generate the virus as described [38]. The infectious clone pACNR-FLYF-17Dx (kindly provided by Charles Rice) was used to generate non-reporter YFV-17D as previously described [39]. VEEV-GFP (strain TC83, a kind gift of Ilya Frolov) was generated by passaging in BHK-21J cells. VSV (kindly provided by Jack Rose) was generated by passaging in BHK-21J cells. ENTV (ATCC VR-378) and RBV (ATCC VR-1263) were produced by passaging in BHK-21J cells. Mengovirus (a kind gift of Julie Pfeiffer), echovirus E11 (a kind gift of Carolyn Coyne), and poliovirus (a kind gift of Julie Pfeiffer) were propagated on HeLa cells. DENV (serotype 2 strain 16681, bearing a L52F mutation in NS4B) was propagated as previously described [40]. HSV-1 (a kind gift of David Leib) was produced by

passaging in VeroE6 cells. Human coronavirus OC43 (ATCC strain VR-1558) was propagated in HCT-8 cells as specified by the ATCC. Viral titers were determined by antibody staining (MAB9012, Millipore) and flow cytometry. All viral stocks were clarified by centrifugation, aliquoted, and stored at -80˚C until use.

## Lentiviral pseudoparticle production and transductions

All lentiviral pseudoparticles were generated by co-transfecting sub-confluent 293T cells with expression plasmid pSCRPSY [41], HIV-1 gag-pol, and VSV-glycoprotein at a ratio of 25:5:1 using XtremeGene 9 (Roche). Two to six hours post-transfection, media was replaced with DMEM containing 3% FBS. Supernatants were collected at 48h and 72h, pooled, either clarified by centrifugation or filtered with a 0.45μM filter, supplemented with 20mM HEPES, aliquoted, and stored at -80˚C until use.

Cells were either transduced by passive infection or by spinoculation. Briefly, lentivirus was added to a minimum volume of transduction media (3% FBS, appropriate base media for each cell line, 4μg/mL polybrene, 20mM HEPES) and added to cells. For passive transductions, cells were allowed to rest with pseudoparticle-containing media for 1–2 hours before addition of complete medium. For spinoculations, cells were centrifuged at 800 x *g* at 37˚C for 40 minutes, after which media was replaced with standard growth media.

## Viral infections

Cells were seeded at 50,000–100,000 (24-well plate) or 4,000,000 (10cm dish) cells per well, depending upon experiment endpoint, the day prior to infection. Virus was added to cells in a minimal volume and incubated for one hour. After incubation, complete media was added to maintain cells until harvest. Unless specifically stated, all infections were performed at an MOI $\leq$ 1 infectious units per cell to ensure that most cells were maximally infected by only one infectious unit. All infected cells were incubated at 37˚C with the exception of HCo-V-OC43, which was incubated at 33˚C. Unless specifically mentioned in the figure legend, infectivity for experiments is quantified by flow cytometry. All WNV infections were performed in a Biosafety Level 3 (BSL3) facility according to institutional guidelines provided by the UT Southwestern Office of Business and Safety Continuity.

## Cloning

Mammalian *RTP4* constructs and vector controls were previously cloned [5,36]. All other RTP homologs were synthesized as gBlocks (IDT). Loss-of-function point mutations were introduced via QuickChange mutagenesis using Herculase II (Agilent). Relevant primers are included in S3 Table.

## Poly(I:C) transfection of A6 cells

One day prior to transfection, A6 cells were seeded at 400,000 cells per well on a 6-well plate in 2mL of media. Cells were transfected with 2μg/well of poly(I:C) (HMW, InvivoGen) using Lipofectamine 3000. Briefly, a master mix containing 2μg of poly(I:C), 4μl of p3000 reagent, and 100μL of OptiMEM was mixed with a master mix containing 100μl of OptiMEM and 6μl of Lipofectamine 3000, incubated for 10 minutes, and added directly to each well. For mock treatments, the same protocol was used with the omission of poly(I:C).

## RNA isolations and sequencing

RNA was isolated using Direct-zol (Zymo). Samples were treated with DNase, per manufacturer recommendation. For RNA sequencing, three biological replicates per treatment group were performed and independently sequenced. RNA Sequencing was performed by Novagene (NovaSeq 6000 PE150 configuration). As input, 1μg RNA per sample was polyA-enriched and sequencing libraries were generated using NEBNext Ultra RNA Library Prep Kit for Illumina (NEB, USA) was used according to manufacturer's recommendations.

## Analysis of RNA-seq data

Reads were mapped to the NCBI Xenopus_Laevis_V2 assembly (GCF_001663975.1) using HISAT2. DESeq2 [42] was used to identify differentially-expressed genes. Adjusted P values were calculated using Benjamini and Hochberg's approach for controlling the False Discovery Rate (FDR). For temporal clustering analysis, DESeq2 was used to filter low-count transcripts (final count: 22,208) and perform pairwise comparisons of treated conditions and mock. $Log_2$-Fold-Change values were subsequently scaled and per-transcript induction profiles were clustered using Ward's method [43] as implemented in R.

## Annotation of *Xenopus laevis* transcripts

BLAST was used to confirm transcript identity by homology. To identify ISGs, homologs were assessed for induction on either the Orthologous Clusters of Interferon-Stimulated Genes database (http://isg.data.cvr.ac.uk/) or the Interferome (interferome.org/).

## Intracellular antibody staining for flow cytometry

Intracellular staining was performed using the CytoFix/Cytoperm Solution Kit (BD). Briefly, cells were fixed/permeabilized for 20 minutes, washed once, incubated in primary antibody (4G2: 1:2500, J2: 1:2000, 542-7D: 1:500) for 22 minutes, washed once, incubated with secondary antibody for 22 minutes, washed once, and resuspended in FACS buffer (3% FBS/PBS). Antibodies raised against the following viral antigens were used for assessing viral infection: E protein (4G2 (BioXCell): DENV, RBV, ENTV); N protein (542-7D (Millipore-Sigma): HCo-V-OC43); dsRNA (J2 (Scicons): PV, E11, MenV).

## Flow cytometry

Samples were run in a Stratedigm S1000 flow cytometer with a A600 96-well plate reader. When necessary, compensation of overlapping fluorescent signal was performed at the time of collection in CellCapture (Stratedigm). FlowJo (BD) was used to quantify data.

## In vitro transcription of viral and replicon RNA

Viral infectious clone and replicon (YFV-R.luc2A-RP) [44] RNA was transcribed using the mMessage mMachine SP6 kit (Ambion). RNA was purified using either RNeasy mini kit (Qiagen) or MEGAClear (ThermoFisher).

## Electroporation of viral RNA

Electroporations were performed as previously described [45]. Briefly, BHK-21J or *STAT1*$^{-/-}$ human fibroblasts were trypsinized, washed in ice-cold PBS, and 8E6 cells in 400ul of PBS were aliquotted into cuvettes along with 5μg of viral RNA. Cells were electroporated at 860V with five pulses and re-seeded into flasks or dishes for production.

## Western blotting

Unless otherwise noted, cells were lysed directly in 1x SDS loading buffer (10% glycerol, 5% BME, 62.5mM TRIS-HCl pH 6.8, 2% SDS, and BPB), boiled, and sonicated (Sonics Vibra-Cell CV188). Samples were run on 12% TGX FastCast acrylamide gels (Bio-Rad) and transferred to nitrocellulose or PVDF membranes using a TransBlot Turbo system (Bio-Rad). Blots were blocked in 5% dry milk/TBS-T (10mM Tris, pH 7.5/50mM NaCl/0.1% Tween-20) for 30 minutes to an hour at RT or overnight at 4˚C. Primary antibodies were diluted in 5%BSA/TBS-T and added for 1 to 2 hours at RT or overnight at 4˚C. Blots were washed four times in TBS-T before addition of HRP-conjugated secondary antibody in 5% milk for thirty minutes. Blots were washed four times in TBS-T prior to detection with either Pierce ECL (ThermoFisher) or Clarity ECL (Bio-Rad) substrate and exposure to radiography film. For some blots, LI-COR IRDye secondary antibodies were used and signal was detected using a LI-COR Odyssey Fc detection system.

## Replicon assay

Cells were seeded at 35,000 cells per well in 48 well plates the day before transfection. 100ng YFRluc-2A RNA was transfected into each well using TransIT-mRNA (Mirus). Cells were washed once with PBS and lysed in Renilla lysis buffer and assayed using the Renilla Luciferase Assay System (Promega).

## CLIP-qPCR

CLIP experiments were performed as previously described with slight modification [46]. Briefly, cells were washed with PBS and cross-linked with 150mJ/cm2 in a Spectrolinker XL1000 (Spectroline). Cells were scraped, pelleted, and snap-frozen. Cells were thawed and lysed in SDS lysis buffer (0.5% SDS, 50mM Tris-Cl pH 6.8, 1mM EDTA, 0.125mg/mL heparin, 2.5mg/mL torula yeast RNA (Sigma), and 1x protease inhibitors (Roche)). Samples were boiled at 65C for 5 minutes and returned to ice. Buffer was adjusted to RIPA (1%NP-40, 0.5% sodium deoxycholate, 0.1% SDS, 150mM NaCl, 50mM Tris-Cl pH 8.0, 2mM EDTA) by addition of a correction buffer (1.25%NP-40, 0.625% sodium deoxycholate, 62.5mM Tris-HCl pH 8.0, 2.25mM EDTA, 187.5mM NaCl). Lysate was passed through a QIAshredder (Qiagen) twice. Lysates were cleared by three high-speed spins with tube transfers. Cleared lysates were supplemented with 5mM CaCl2 and treated with 30U of DNase (NEB) for 15 minutes. Antibody conjugated beads (preconjugated HA, Pierce) were added to samples. Samples were rotated end over end at 4˚C for 2h. Samples were placed on a magnetic separator and washed three times with RIPA, once with RIPA supplemented with 1M Urea, and twice with RIPA. RNA was eluted from beads by addition of Proteinase K buffer (0.5mg/mL Proteinase K (Ambion), 0.5% SDS, 20mM Tris-Cl pH 7.5, 5mM EDTA, 16.7ng/µl GlycoBlue (Invitrogen), 0.1mg/mL torula yeast RNA) and incubation for 1-2h with shaking at 37˚C. Following elution, RNA was extracted with phenol-chloroform-isoamyl alcohol, extracted with chloroform, and precipitated. For some experiments, RT-qPCR was directly performed using OneStep RT-PCR (Qiagen). For others, RNA was subsequently DNase-treated, re-purified, and cDNA was generated using Superscript IV and random hexamers. cDNA was treated with RNase H and RNase A, precipitated, and resuspended in a low volume of water for storage at -20˚C. cDNA was diluted prior to qPCR.

## Phylogenetic analysis

Multiple sequence alignments were generated using MUSCLE as implemented in MEGA X [47]. Alignments were trimmed using Gblocks [48]. When required, file formats were

converted using ALTER [49]. PAML was used to assess signatures of evolutionary pressure present in nucleic acid alignments. Briefly, CodeML was used with the F3x4 codon frequency table and default settings [17]. Likelihood ratio tests were used to compare Model 8 (beta and omega—allowing for positive selection) and Model 7 (beta—no positive selection), as well as Model 8 and Model 8a (beta and omega, constrained to no positive selection). Sites that passed a stringent test (Bayes empirical Bayes) test were considered to be undergoing positive selection. To perform a free-ratio analysis, PAML was run with Model 1 (branch) and NSsites = 0 to obtain a dN/dS value for each branch. HyPhy analyses (FUBAR, BUSTED, aBSREL, GARD) were either run on a local system or performed as implemented on DataMonkey [50].

## Synteny analysis

NCBI and EMSEMBL genome viewers were used to manually assess the location of RTPs relative to proximal genomic elements. NCBI BLAST was used to probe for pseudogenes or unannotated RTPs.

## Statistical analysis

Statistical analyses were performed using GraphPad Prism unless otherwise noted. Unless otherwise indicated, all comparisons are relative to control (Ctrl), as labeled. For data with two groups, two-tailed t-tests were used. For data with more than two groups, ANOVA tests were used and appropriate corrections were made for multiple hypothesis testing. Additional details are available in all figure legends where any statistical tests were performed. Unless otherwise specified, P values are denoted as follows: n.s. not significant, $*P<0.05$, $**P<0.01$, $***P<0.001$, $****P<0.0001$.

## Supporting information

**S1 Fig.** A)Collapsed maximum-likelihood trees for N-terminal and C-terminal fish RTP gene fragments, relative to a predicted recombination breakpoint (nt383). Numbers indicate the percentage of 50 bootstrap replicates for which the associated taxa clustered together. Nodes with <50% bootstrap confidence were collapsed. Trees were generated as in Fig 1B. (TIF)

**S2 Fig.** A) Huh7.5 cells ectopically expressing the indicate RTPs or a vector control were infected with a panel of viruses at an MOI of 0.5 to 1. Cells were harvested after the completion of approximately one replication cycle and percent infection was assessed by flow cytometry. Cells represent the mean of n = 2 (HCV) or n = 3 (all other viruses) biological replicates, normalized to control. B) Western blot of ectopically-expressed RTPs. For each screen replicate, an additional well of cells was harvested alongside infections for protein expression analysis. Blot is representative of n = 3 replicates. C) Representative micrograph of paRTP3-expressing Huh7.5 cells and vector control cells. Scale bar: 20μm. D) Huh7.5 cells expressing the indicated construct were seeded at 10,000 cells per well on a 96-well plate one day prior to assay. The day of WST assay, WST reagent (Takara) was added (10uL in 100uL media) and cells were incubated for two hours, after which visual absorbance was assayed. In parallel, cell density was measured using Cell Titer-Glo (Promega) and luminometry. (TIF)

**S3 Fig.** A) Western blot of ectopically-expressed RTPs. Vertical dotted line indicates splice site for two independently-run gels. For each screen replicate, an additional well of cells was harvested alongside infections for protein expression analysis. Blot is representative of n = 2 replicates. B) Raw data related to Fig 4A. Huh7.5 cells ectopically expressing the indicate RTPs

or a vector control were infected with a panel of RNA viruses at an MOI of 0.5 to 1. Cells were harvested after the completion of approximately one viral life cycle and percent infection was assessed by flow cytometry. Bars represent the mean ± SD of n = 2 biological replicates. C) Cartoon representation of *X. laevis* RTPαL1 and RTPγS with paRTP4 as a reference. Yellow boxes denote N-terminal CXXC motifs, the third of which was targeted for disruption by site-directed mutagenesis to generate *ZFD RTPs. D) Western blot of ectopically-expressed RTPγS and RTPγS*ZFD. Blot is representative of n = 2 replicates. E) Huh7.5 cells expressing HA. RTPαL1 or a vector control were infected with MenV (MOI of 2) for 6h. CLIP-qPCR identified RNA bound by HA.RTPαL1. UV: UV crosslinked HA.RTPαL1 cells. NoUV: non-crosslinked HA.RTPαL1 cells. Vector: UV crosslinked vector control cells. Bars: mean ± SD of n = 3 biological replicates.
(TIF)

**S1 Table. aBSREL results for *X. laevis* RTPs.**
(XLSX)

**S2 Table. RNA sequencing results.** Includes differentially-expressed genes and annotated lists of overall top-upregulated transcripts, as well as those in SC4 and SC5.
(XLSX)

**S3 Table. List of oligos used in this study.**
(XLSX)

**S4 Table. Accession numbers for RTP sequences used in this study.**
(XLSX)

**S5 Table. Underlying numerical data generated by this study.**
(XLSX)

**S1 File. Zip folder containing alignments and ML tree of RTPs used in this study.** File A) Alignment of RTP sequences used to generate Fig 1A. File B) Tree depicted in 1A, with species and gene names indicated. File C) Alignment of RTP sequences used in Fig 1B. File D) Alignments used for phylogenetic analysis in Table 1. File E) Alignment of RTP sequences used to generate Fig 2C.
(ZIP)

## Acknowledgments

We thank Jennifer Eitson and Wenchun Fan for technical assistance in preparing virus stocks.

## Author Contributions

**Conceptualization:** Ian N. Boys, John W. Schoggins.

**Formal analysis:** Ian N. Boys.

**Funding acquisition:** John W. Schoggins.

**Investigation:** Ian N. Boys.

**Resources:** Ian N. Boys, Katrina B. Mar.

**Supervision:** John W. Schoggins.

**Visualization:** Ian N. Boys.

**Writing – original draft:** Ian N. Boys.

**Writing – review & editing:** Ian N. Boys, Katrina B. Mar, John W. Schoggins.

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
