## [Decision Letter · Decision Letter 0]

14 Apr 2021

Dear Dr Schoggins,

Thank you very much for submitting your Research Article entitled 'Functional-genomic analysis reveals intraspecies diversification of antiviral receptor transporter proteins in Xenopus laevis' to PLOS Genetics.

The manuscript was fully evaluated at the editorial level and by independent peer reviewers. The reviewers appreciated the attention to an important topic but identified some concerns that we ask you address in a revised manuscript

We therefore ask you to modify the manuscript according to the review recommendations. Your revisions should address the specific points made by each reviewer.

[LINK]

Yours sincerely,

Takashi Gojobori

Associate Editor

PLOS Genetics

Hua Tang

Section Editor: Natural Variation

PLOS Genetics

Reviewer's Responses to Questions

**Comments to the Authors:**

Reviewer #1: PGENETICS-D-21-00228

This manuscript describes the evolutionary comparison of receptor transporter proteins (RTPs) and focuses on the diverged and expanded evolution of these in X. laevis. The authors underline the expansion of the Xenopus RTPs, show that RTPs from several species are upregulated in response to immune challenge and that some of them are important during RNA virus infections.

Please find my comments and suggestions below.

In your introduction, please elaborate on the mechanism by which RTP4 restricts viral replication.

Fig. 1 is very difficult to read. Please consider increasing the resolution in A and B and increasing the font or somehow highlighting key species in A.

The synteny analyses in Fig. 2A is difficult to discern. Please clearly indicate the respective RTP genes including numbers and label several of the neighboring genes. Please also indicate the s and L allelic forms of these genes in X. laevis.

There is a gap between pg. 6 and 7. Possibly formatting?

Reviewer #2: In this manuscript, the authors perform an evolutionary analysis of Receptor Transporter Proteins (RTPs) in vertebrates. RTP4 in humans is an interferon stimulated gene and was previously shown by this group to have anti-flavivirus activity. Here, they identified signs of positive selection of RTPs in several species, which often indicates virus-host conflicts driving accelerated evolution. They further identified expansions of RTPs in certain species, including xenopus laevis, which encodes 11 RTPs. They went on to provide the first RNA seq timecourse of xenopus cells stimulated with the viral mimic polyI:C. This is a valuable dataset in and of itself, and this identified xenopus RTPs that were upregulated by polyI:C stimulation. They then ectopically expressed several xenopus RTPs in human cells and discovered antiviral activities against specific viruses or virus families, including flaviviruses and picornaviruses. These data provide some of the first evidence that antiviral activity of RTPs is present outside of mammals and is thus evolutionarily more ancient than previously appreciated. Overall, I find this paper to reach the threshold of both significance and quality for publication in PLOS Genetics with only minor text additions as noted below.

1. While the authors appropriately mention certain limitations of their study, one limitation that was not mentioned was the lack of infection or RTP manipulation in xenopus cells. If this is technically challenging, a brief discussion of this would be helpful.

2. It would be nice if the authors could present the data in the supplementary tables in a more user-friendly way with captions and table headings. For example, in Supp Table 2, a simple list of the upregulated xenopus gene names and fold upregulation at each timepoint would make the dataset more accessible for quick inspection by the average reader who may want to quickly scan for a specific ISG.

Reviewer #3: This is a very thorough and well performed study where authors describe the immunogenetics of antiviral RTPs in amphibians. The team reports an expansion of these molecules in Xenopus (11 RTPs) and perform in-depth phylogenetic analyses to ascertain the evolutionary history of RTPs in amniotes and anamniotes. They additionally analyze differences between the Xenopus S and L genomes. The authors also present a series of in vitro assays using an ectopic expression system to show responses to flaviviruses and in vitro transfection of Xenopus A6 cells with poly I:C. I enjoyed the CLIP experiments that point towards similar antiviral mechanisms of Xenopus RPL and mammalian RPL4.

The manuscript is well written and the experiments include the correct controls.

I have two main questions:

1- Where are RTPs expressed in X. laevis (what tissues) at the steady state? Are they constitutively expressed? Is there high expression in the olfactory system as mentioned at the end of the discussion compared to lymphoid tissues?

2.- Authors state the main limitation of this study in lines 252-255 of the discussion. Given all the published RNA-Seq datasets of amphibians infected with amphibian-relevant pathogens such as ranavirus and Bd, I highly recommend that the authors complement their in vitro findings with in vivo data that has already been published. Data mining these datasets will help support the idea that amphibian RLPs are indeed antiviral in a more biologically relevant model. Further, having the comparison between Bd versus ranavirus datasets would be really interesting to ascertain antiviral specific roles.

Minor:

- RNA-Seq data: I cannot find the number of replicates per treatment group in the materials and methods

- Phylogenetic trees: I cannot find the accession numbers of the sequences for all the RTP molecules used in Figure 1.

**Have all data underlying the figures and results presented in the manuscript been provided?**

Reviewer #1: Yes

Reviewer #2: **No: **I don't think I saw all of the underlying data for bar graphs present in any of the supplementary tables, though I'm not sure this is necessary since their bar graphs include overlay of individual data points.

Reviewer #3: Yes

PLOS authors have the option to publish the peer review history of their article (what does this mean?). If published, this will include your full peer review and any attached files.

Reviewer #1: No

Reviewer #2: **Yes: **Jacob Yount

Reviewer #3: **Yes: **Dr. Irene Salinas

---

## [Editor Report · Decision Letter 1]

4 May 2021

Dear Dr Schoggins,

We are pleased to inform you that your manuscript entitled "Functional-genomic analysis reveals intraspecies diversification of antiviral receptor transporter proteins in Xenopus laevis" has been editorially accepted for publication in PLOS Genetics. Congratulations!

Yours sincerely,

Takashi Gojobori

Associate Editor

PLOS Genetics

Hua Tang

Section Editor: Natural Variation

PLOS Genetics

Comments from the reviewers (if applicable):

**Data Deposition**

http://datadryad.org/submit?journalID=pgenetics&manu=PGENETICS-D-21-00228R1

**Press Queries**

---

## [Editor Report · Acceptance letter]

17 May 2021

PGENETICS-D-21-00228R1 

Functional-genomic analysis reveals intraspecies diversification of antiviral receptor transporter proteins in Xenopus laevis 

Dear Dr Schoggins, 

We are pleased to inform you that your manuscript entitled "Functional-genomic analysis reveals intraspecies diversification of antiviral receptor transporter proteins in Xenopus laevis" has been formally accepted for publication in PLOS Genetics! Your manuscript is now with our production department and you will be notified of the publication date in due course.

With kind regards,

Katalin Szabo

PLOS Genetics

On behalf of:
